# H2RBox: Horizontal Box Annotation is All You Need for Oriented Object Detection

**Xue Yang[1], Gefan Zhang[1,2], Wentong Li[3], Xuehui Wang[1], Yue Zhou[1], Junchi Yan[1,4,*]**
[1]MoE Key Lab of Artificial Intelligence, Shanghai Jiao Tong University
[2]COWAROBOT Co. Ltd.    [3]Zhejiang University    [4]Shanghai AI Laboratory
`{yangxue-2019-sjtu,lizaozhouke}@sjtu.edu.cn, liwentong@zju.edu.cn`
`{wangxuehui,sjtu_zy,yanjunchi}@sjtu.edu.cn`
Jittor Code: https://github.com/yangxue0827/h2rbox-jittor
PyTorch Code: https://github.com/yangxue0827/h2rbox-mmrotate

## Abstract

Oriented object detection emerges in many applications from aerial images to autonomous driving, while many existing detection benchmarks are annotated with horizontal bounding box only which is also less costive than fine-grained rotated box, leading to a gap between the readily available training corpus and the rising demand for oriented object detection. This paper proposes a simple yet effective oriented object detection approach called H2RBox merely using horizontal box annotation for weakly-supervised training, which closes the above gap and shows competitive performance even against those trained with rotated boxes. The cores of our method are weakly- and self-supervised learning, which predicts the angle of the object by learning the consistency of two different views. To our best knowledge, H2RBox is the first horizontal box annotation-based oriented object detector. Compared to an alternative i.e. horizontal box-supervised instance segmentation with our post adaption to oriented object detection, our approach is not susceptible to the prediction quality of mask and can perform more robustly in complex scenes containing a large number of dense objects and outliers. Experimental results show that H2RBox has significant performance and speed advantages over horizontal box-supervised instance segmentation methods, as well as lower memory requirements. While compared to rotated box-supervised oriented object detectors, our method shows very close performance and speed. The source code is available at PyTorch-based MMRotate and Jittor-based JDet.

## 1 Introduction

In addition to the relatively matured area of horizontal object detection (Liu et al., 2020), oriented object detection has received extensive attention, especially for complex scenes, whereby fine-grained bounding box (e.g. rotated/quadrilateral bounding box) is needed, e.g. aerial images (Ding et al., 2019; Yang et al., 2019a), scene text (Zhou et al., 2017), retail scenes (Pan et al., 2020) etc.

Despite the increasing popularity of oriented object detection, many existing datasets are annotated with horizontal boxes (HBox) which may not be compatible (at least on the surface) for training an oriented detector. Hence labor-intensive re-annotation[1] have been performed on existing horizontal-annotated datasets. For example, DIOR-R (Cheng et al., 2022) and SKU110K-R (Pan et al., 2020) are rotated box (RBox) annotations of the aerial image dataset DIOR (192K instances) (Li et al., 2020) and the retail scene SKU110K (1,733K instances) (Goldman et al., 2019), respectively.

One attractive question arises that if one can achieve weakly supervised learning for oriented object detection by only using (the more readily available) HBox annotations than RBox ones. One poten-

---

*Correspondence author is Junchi Yan. The work was in part supported by National Key Research and Development Program of China (2020AAA0107600), National Natural Science Foundation of China (62222607), and Shanghai Municipal Science and Technology Major Project (2021SHZDZX0102).

[1]The annotation cost (in price) of the RBox is about 36.5% ($86 vs. $63) higher than that of the HBox according to https://cloud.google.com/ai-platform/data-labeling/pricing.

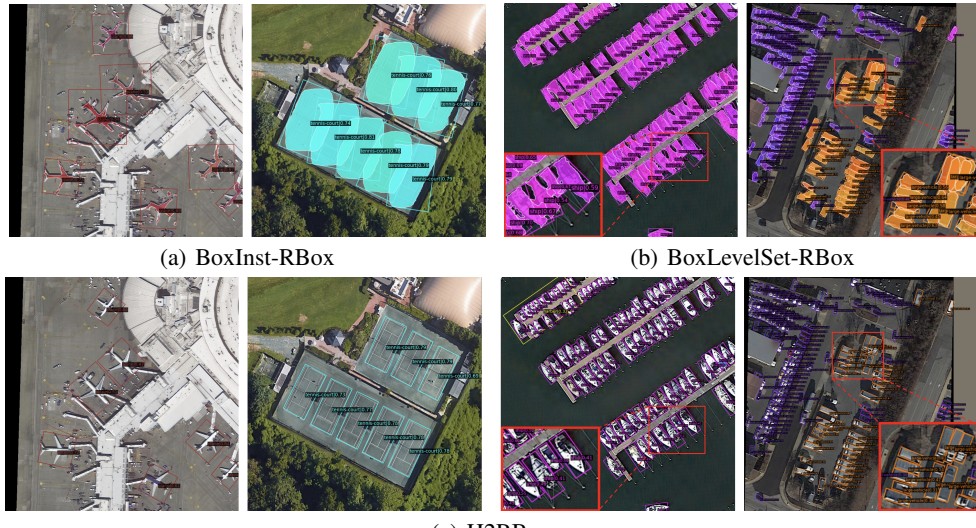

Figure 1: Visual comparison of three HBox-supervised rotated detectors on aircraft detection (Wei et al., 2020), ship detection (Yang et al., 2018), vehicle detection (Azimi et al., 2021), etc. The HBox-Mask-RBox style methods, i.e. BoxInst-RBox (Tian et al., 2021) and BoxLevelSet-RBox (Li et al., 2022b), perform not well in complex and object-cluttered scenes.

tial and verified technique in our experiments is HBox-supervised instance segmentation, concerning with BoxInst (Tian et al., 2021), BoxLevelSet (Li et al., 2022b), etc. Based on the segmentation mask by these methods, one can readily obtain the final RBox by finding its minimum circumscribed rectangle, and we term the above procedure as HBox-Mask-RBox style methods i.e. BoxInst-RBox and BoxLevelSet-RBox in this paper. Yet it in fact involves a potentially more challenging task i.e. instance segmentation whose quality can be sensitive to the background noise, and it can influence heavily on the subsequent RBox detection step, especially given complex scenes (in Fig. 1(a)) and the objects are crowded (in Fig. 1(b)). Also, involving segmentation is often more computational costive and the whole procedure can be time consuming (see Tab. 1-2).

In this paper, we propose a simple yet effective approach, dubbed as **HBox-to-RBox (H2RBox)**, which achieves close performance to those RBox annotation supervised methods e.g. (Han et al., 2021b; Yang et al., 2023a) by only using HBox annotations, and even outperforms in considerable amount of cases as shown in our experiments. The cores of our method are weakly- and self-supervised learning, which predicts the angle of the object by learning the enforced consistency between two different views. Specifically, we predict five offsets in the regression sub-network based on FCOS (Tian et al., 2019) in the WS branch (see Fig. 2 left) so that the final decoded outputs are RBoxes. Since we only have horizontal box annotations, we use the horizontal circumscribed rectangle of the predicted RBox when computing the regression loss. Ideally, predicted RBoxes and corresponding ground truth (GT) RBoxes (unlabeled) have highly overlapping horizontal circumscribed rectangles. In the SS branch (see Fig. 2 right), we rotate the input image by a randomly angle and predict the corresponding RBox through a regression sub-network. Then, the consistency of RBoxes between the two branches, including scale consistency and spatial location consistency, are learned to eliminate the undesired cases to ensure the reliability of the WS branch. Our main contributions are as follows:

**1)** To our best knowledge, we propose the first HBox annotation-based oriented object detector. Specifically, a weakly- and self-supervised angle learning paradigm is devised which closes the gap between HBox training and RBox testing, and it can serve as a plugin for existing detectors.

**2)** We prove through geometric equations that the predicted RBox is the correct GT RBox under our designed pipeline and consistency loss, and does not rely on not-fully-verified/ad-hoc assumptions, e.g. color-pairwise affinity in BoxInst or additional intermediate results whose quality cannot be ensured, e.g. feature map used by many weakly supervised methods (Wang et al., 2022).

**3)** Compared with the potential alternatives e.g. HBox-Mask-RBox whose instance segmentation part is fulfilled by the state-of-the-art BoxInst, our H2RBox outperforms by about 14% mAP

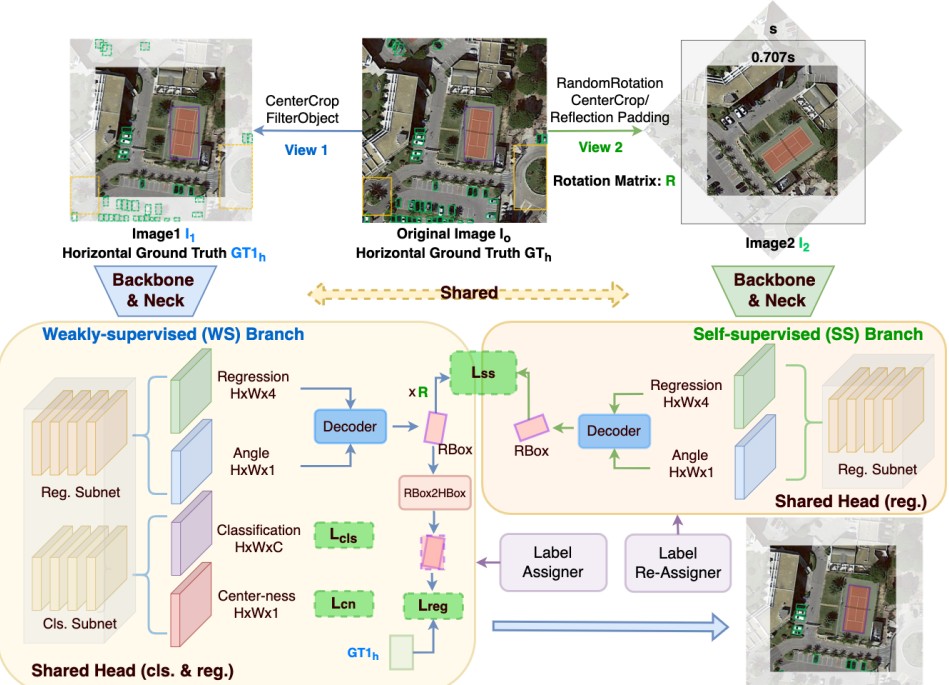

Figure 2: Our H2RBox consists of two branches respectively fed with two augmented views (**View 1 and View 2**) of the input image. The left **Weakly-supervised Branch** in general can be any rotated object detector (FCOS here) for RBox prediction, whose circumscribed HBox is used for supervised learning given the GT HBox label in the sense of weakly-supervised learning. This branch is also used for test-stage inference. The right **Self-supervised Branch** tires to achieve RBox prediction consistency of the two views with self-supervised learning. Image is from the DIOR-R dataset.

(67.90% vs. 53.59%) on DOTA-v1.0 dataset, requiring only one third of its computational resources (6.25 GB vs. 19.93 GB), and being around $12\times$ faster in inference (31.6 fps vs. 2.7 fps).

**4)** Compared with the fully RBox annotation-supervised rotation detector FCOS, H2RBox is only 0.91% (74.40% vs. 75.31%) and 1.01% (33.15% vs. 34.16%) behind on DOTA-v1.0 and DIOR-R, respectively. Furthermore, we do not add extra computation in the inference stage, thus maintaining a comparable detection speed, about 29.1 FPS vs. 29.5 FPS on DOTA-v1.0.

## 2 RELATED WORK

**RBox-supervised Oriented Object Detection.** Oriented object detection in visual images has received increasing attention across different areas e.g. aerial image (Xu et al., 2020; Yang et al., 2022; 2023a; Hou et al., 2023), scene text (Zhou et al., 2017; Liao et al., 2018), retail (Pan et al., 2020; Chen et al., 2020), etc. Earlier methods including RRPN (Ma et al., 2018), ROI-Transformer (Ding et al., 2019) and ReDet (Han et al., 2021b) directly perform angle regression. To address the loss discontinuity and regression inconsistency due to periodicity of angle, subsequent works either convert the parameterization of the rotated bounding box into 2-D Gaussian distributions (Yang et al., 2021c;d) or transform the angle regression to classification (Yang et al., 2021a; Yang & Yan, 2022). (Hou et al., 2022; Li et al., 2022a) introduce the adaptive point set for object representation to mitigate the angle regression sensitivity and meanwhile captures instances' semantic information.

**HBox-supervised Instance Segmentation and Its Potential for Oriented Object Detection.** The bold idea of purely using HBox-annotations to train a rotated object detector is attractive yet still rarely studied in literature, which can be seen as a weakly-supervised (WS) learning paradigm for oriented object detection. A related and better-studied technique is HBox-supervised instance segmentation, which tries to segment instance based on the HBox annotations for WS training. For instance, SDI (Khoreva et al., 2017) relies on the region proposals generated by MCG (Pont-Tuset

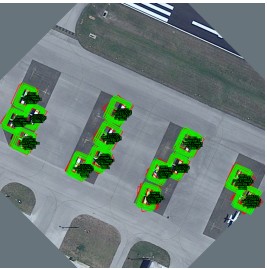 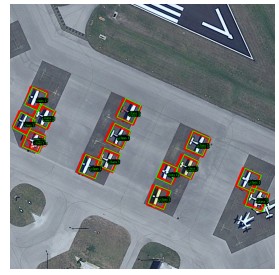 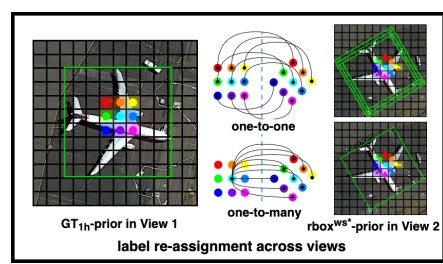

(a) O2O by zeros padding.  (b) O2M by reflect padding.    (c) The two re-assignment strategies.

Figure 3: Comparison of different padding methods (Sec. 3.1) and re-assignment strategies (Sec. 3.4). Green and red RBox represent the target $rbox^{ws*}$ and $rbox^{ss}$, respectively.

et al., 2016) and uses an iterative training process to refine the segmentation. BBTP (Hsu et al., 2019) formulates the HBox-supervised instance segmentation into a multiple instance learning problem based on Mask R-CNN (He et al., 2017). BoxInst (Tian et al., 2021) uses the color-pairwise affinity with box constraint under an efficient RoI-free CondInst (Tian et al., 2020). BoxLevelSet (Li et al., 2022b) introduces an energy function to predict the instance-aware mask as the level set.

Though one can obtain the final object orientation by certain means based on the segmentation mask from the above instance segmentation methods, e.g. by finding the minimum circumscribed rectangle, we argue and show in our experiments that such an HBox-Mask-RBox pipeline can be complex (segmentation can be even more difficult than rotation detection – see Fig. 1) and expensive in the presence of dense objects and background noises. Hence we aim to skip the segmentation step and build an HBox-to-RBox paradigm which has not been studied before to our best knowledge.

## 3 PROPOSED METHOD

The overview of the H2RBox is shown in Fig. 2. Two augmented views are generated and information leakage is avoided for training overfitting. There are two branches. One branch is used for weakly-supervised (WS) learning where the supervision is the GT HBox from the training data, and the regression loss is calculated between the circumscribed HBox derived from the predicted RBox by this branch and GT HBox. The other branch is trained by self-supervised (SS) learning that involves two augmented views of the raw input image, which encourages to obtain the consistent RBox prediction between the two views. The final loss is the weighted sum of the WS loss and SS loss. Note that the test-stage prediction is concerned only with the WS branch.

### 3.1 AUGMENTED VIEW GENERATION

In line with the general idea of self-supervised learning by data augmentation, given the input image, we perform random rotation to generate View 2 while keeping View 1 consistent with the input image, as shown in Fig. 2. However, rotation transformation will geometrically and inevitably introduce an artificial black border area and leads to the risk of GT angle information leakage. We provide two available techniques to resolve this issue:

1) Center Region Cropping: Crop a $\frac{\sqrt{2}}{2}s \times \frac{\sqrt{2}}{2}s$ area[2] in the center of the image.

2) Reflection Padding: Fill the black border area by reflection padding.

If the Center Region Cropping is used in View 2, View 1 also needs to perform the same operation and filter the corresponding ground truth. In contrast, Reflection Padding works better than Center Region Cropping because it preserves as much of the area as possible while maintaining a higher image resolution. Fig. 3(a) and Fig. 3(b) compare zeros padding and reflection padding. Note that the black border area does not participate in the regression loss calculation in the SS branch, so it does not matter that this region is filled with unlabeled foreground objects by reflection padding.

---

[2]When the rotation angle is a multiple of $45°$, the black border area reaches its peak, so the side length of the largest crop area is $\frac{\sqrt{2}}{2}$ of the side length of the original image ($s$), refer to the View 2 in Fig. 2.

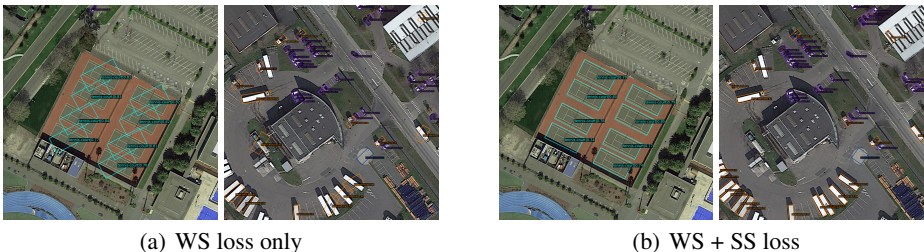

(a) WS loss only  (b) WS + SS loss

Figure 4: Visual comparison of our methods with and without the SS loss used in the SS branch. It can help learn the scale and spatial location consistency between the two branches.

## 3.2 THE WEAKLY-SUPERVISED (WS) BRANCH

The two generated views (**View 1** and **View 2**) are respectively fed into the two branches with the parameter-shared backbone and neck, specified as ResNet (He et al., 2016) and FPN (Lin et al., 2017a) as shown in Fig. 2. The WS branch here is specified by a FCOS-based rotated object detector, as involved for both training and inference. This branch contains regression and classification sub-networks to predict RBox, category, and center-ness. Recall that we can not use the predicted RBox to calculate the final regression directly as there is no RBox annotation but HBox only. Therefore, we first convert the predicted RBox into the corresponding minimum horizontal circumscribed rectangle, for calculating the regression loss

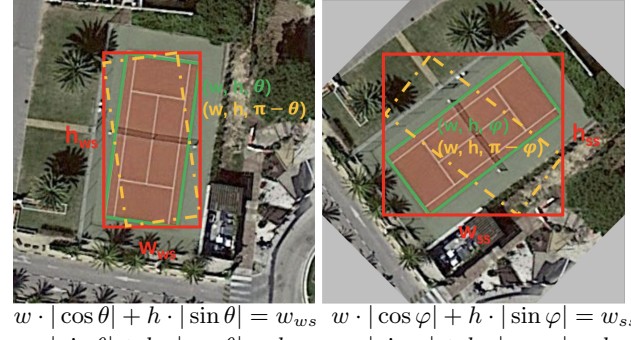

$w \cdot |\cos\theta| + h \cdot |\sin\theta| = w_{ws}$    $w \cdot |\cos\varphi| + h \cdot |\sin\varphi| = w_{ss}$
$w \cdot |\sin\theta| + h \cdot |\cos\theta| = h_{ws}$    $w \cdot |\sin\varphi| + h \cdot |\cos\varphi| = h_{ss}$

Figure 5: Proof of the relationship between predicted RBox and GT RBox under horizontal circumscribed rectangle constraint and scale constraint. Green and orange RBoxes represent correct coincident prediction $B^c$ and undesired symmetric prediction $B^s$.

between the derived HBox and the GT Hbox annotation (we defer the details of the loss formulation to Sec. 3.5). As the network is better trained, an indirect connection (horizontal circumscribed rectangle constraint) occurs between predicted RBox and GT RBox (unlabeled): *No matter how an object is rotated, their corresponding horizontal circumscribed rectangles are always highly overlapping.* However, as shown in Fig. 4(a), only using WS loss can only localize the objects, while still not effective enough for accurate rotation estimation.

## 3.3 THE SELF-SUPERVISED (SS) BRANCH

As complementary to the WS loss, we further introduce the SS loss. The SS branch only contains one regression sub-network for predicting RBox in the rotated View 2. Given a (random) rotation transformation $\mathbf{R}$ (with degree $\Delta\theta$) as adopted in View 2, the relationship between location $(x, y)$ of View 1 in the WS branch and location $(x^*, y^*)$ of View 2 with rotation $\mathbf{R}$ in the SS branch is:

$$(x^*, y^*) = (x - x_c, y - y_c)\mathbf{R}^\top + (x_c, y_c), \quad \mathbf{R} = \begin{pmatrix} \cos\Delta\theta & -\sin\Delta\theta \\ \sin\Delta\theta & \cos\Delta\theta \end{pmatrix} \quad (1)$$

where $(x_c, y_c)$ is the rotation center (i.e. image center). Recall the label of the black border area (in Fig. 3) in the SS branch is set as invalid and negative samples, which will not participate in the subsequent losses designed below.

Specifically, a scale loss $L_{wh}$ accounts for the scale consistency to enhance the indirect connection described above: *For augmented objects obtained from the same object through different rotations, a set of RBoxes of the same scale are predicted by the detector, and these predicted RBoxes and corresponding GT RBoxes (unlabeled) shall have highly overlapping horizontal circumscribed rectangles.* With such an enhanced indirect connection, including horizontal circumscribed rectangle

constraint and scale constraint, we can limit the prediction results to a limited number of feasible cases, explained as follows:

Fig. 5 shows two cases based on the above enhanced indirect connection, and lists four different expressions for the four variables $(w, h, \theta, \varphi)$. Due to the periodicity of the angles, there are only two feasible solutions to the four equations within the angle definition, i.e. the green GT RBox and the orange symmetric RBox. In other words, with such a strengthened indirect connection, the relationship between predicted RBox and GT RBox is coincident $B^c(w, h, \theta)$ or symmetrical about the center of the object $B^s(w, h, \pi - \theta)$. It can be seen from Fig. 4(a) that there are still many bad cases with extremely inaccurate angles after using $L_{wh}$. Interestingly, if we make a symmetry transformation of these bad cases with their center point, the result becomes much better. When generating views, a geometric prior can be obtained, that is, the spatial transformation relationship between the two views, denoted as $\mathbf{R}$ in Eq. 1. Thus, we can get the following four transformation relationships, marked as $T\langle B_{ws}, B_{ss}\rangle$, between the two branches:

$$T\langle B_{ws}^c, B_{ss}^c\rangle = \{\mathbf{R}\}, \quad T\langle B_{ws}^c, B_{ss}^s\rangle = \{\mathbf{R}, \mathbf{S}\} = \{\mathbf{S}, \mathbf{R}^\top\}$$
$$T\langle B_{ws}^s, B_{ss}^s\rangle = \{\mathbf{R}^\top\}, \quad T\langle B_{ws}^s, B_{ss}^c\rangle = \{\mathbf{R}^\top, \mathbf{S}\} = \{\mathbf{S}, \mathbf{R}\}$$
(2)

where $B_{ws}^c$ and $B_{ss}^s$ represent the coincident bounding box predicted in WS branch and the symmetric bounding box predicted in SS branch, respectively. Here $\mathbf{S}$ denotes symmetric transformation. Take $T\langle B_{ws}^c, B_{ss}^s\rangle = \{\mathbf{R}, \mathbf{S}\}$ as an example, it means $B_{ss}^s = \mathbf{S}(\mathbf{R} \cdot B_{ws}^c)$.

Therefore, an effective way to eliminate the symmetric case is to let the model know that the relationship between the RBoxes predicted by the two branches can only be $\mathbf{R}$. Inspired by above analysis, spatial location loss is used to construct the spatial transformation relationship $\mathbf{R}$ of RBoxes predicted by two branches. Specifically, the RBox predicted by WS branch is first transformed by $\mathbf{R}$, and then several losses (e.g. center point loss $L_{xy}$ and angle loss $L_\theta$) are used to measure its location consistency with the RBox predicted by SS branch. In fact, the spatial location consistency, especially the angle loss, provides a fifth angle constraint equation ($\varphi - \theta = \Delta\theta \neq 0$) so that the system of equations in Fig. 5 have a unique solution (i.e. the predicted RBox is the GT RBox) with non-strict proof, because system of equations are nonlinear. The final SS learning consists of scale-consistent and spatial-location-consistent learning:

$$Sim\langle \mathbf{R} \cdot B_{ws}, B_{ss}\rangle = 1$$
(3)

Fig. 4(b) shows the visualization by using the SS loss, with accurate predictions. The appendix shows visualizations of feasible solutions for different combinations of constraints.

## 3.4 LABEL RE-ASSIGNER

Since the consistency of the prediction results of the two branches needs to be calculated, the labels need to be re-assigned in the SS branch. Specifically, the labels at the location $(x^*, y^*)$ of the SS branch, including center-ness $(cn^*)$, target category $(c^*)$ and target GT HBox $(gtbox^{h*})$, are the same as in the location $(x, y)$ of the WS branch. Besides, we also need to assign the $rbox^{ws}(x_{ws}, y_{ws}, w_{ws}, h_{ws}, \theta_{ws})$ predicted by the WS branch as the target RBox of the SS branch to calculate the SS loss. We propose two reassignment strategies:

**1) One-to-one (O2O) assignment:** With $cn$, $c$ and $gtbox^h$, the $rbox^{ws}$ predicted at location $(x, y)$ in the WS branch is used as the target RBox at $(x^*, y^*)$ of the SS branch (see Fig. 3(a)).

**2) One-to-many (O2M) assignment:** Use the $rbox^{ws}$ closest to the center point of the $gtbox^h_{(x,y)}$ as the target RBox at location $(x^*, y^*)$ of SS branch, as shown in Fig. 3(b).

Fig. 3(c) visualizes the difference between the two re-assignment strategies. After re-assigning, we need to perform an rotation transformation on the $rbox^{ws}$ to get the $rbox^{ws*}(x_{ws}^*, y_{ws}^*, w_{ws}^*, h_{ws}^*, \theta_{ws}^*)$ for calculating the SS loss according to Eq. 3:

$$(x_{ws}^*, y_{ws}^*) = (x_{ws} - x_c, y_{ws} - y_c)\mathbf{R}^\top + (x_c, y_c), \quad (w_{ws}^*, h_{ws}^*) = (w_{ws}, h_{ws}), \quad \theta_{ws}^* = \theta_{ws} + \Delta\theta \quad (4)$$

The the visualized label assignment in Fig. 3 further shows that the SS loss effectively eliminates prediction of the undesired case. The label reassignment of different detectors may require different strategies. The key is to design a suitable matching strategy for the prediction results of the two views, which can allow the network to learn the consistency better.

## 3.5 THE OVERALL LOSS BY COMBINING THE WS AND SS LOSSES

Since the WS branch is a rotated object detector based on FCOS, the losses in this part mainly include the regression $L_{reg}$, classification $L_{cls}$, and center-ness $L_{cn}$. We define the WS loss in the WS branch as follows:

$$L_{ws} = \frac{\mu_1}{N_{pos}} \sum_{(x,y)} L_{cls}(p_{(x,y)}, c_{(x,y)}) + \frac{\mu_2}{N_{pos}} \sum_{(x,y)} L_{cn}(cn'_{(x,y)}, cn_{(x,y)})$$
$$+ \frac{\mu_3}{\sum cn_{pos}} \sum_{(x,y)} \mathbb{1}_{\{c_{(x,y)}>0\}} cn_{(x,y)} L_{reg}\left(r2h(rbox^{ws}_{(x,y)}), gtbox^{h}_{(x,y)}\right) \tag{5}$$

where $L_{cls}$ is the focal loss (Lin et al., 2017b), $L_{cn}$ is cross-entropy loss, and $L_{reg}$ is IoU loss (Yu et al., 2016). $N_{pos}$ denotes the number of positive samples. $p$ and $c$ denote the probability distribution of various classes calculated by Sigmoid function and target category. $rbox^{ws}$ and $gtbox^{h}$ represent the predicted RBox in the WS branch and horizontal GT box, respectively. $cn'$ and $cn$ indicate the predicted and target center-ness. $\mathbb{1}_{\{c_{(x,y)}>0\}}$ is the indicator function, being 1 if $c_{(x,y)} > 0$ and 0 otherwise. The $r2h(\cdot)$ function converts the RBox to its corresponding horizontal circumscribed rectangle. We set the hyperparameters $\mu_1 = 1$, $\mu_2 = 1$ and $\mu_3 = 1$ by default.

Then, the SS loss between $rbox^{ws*}(x^*_{ws}, y^*_{ws}, w^*_{ws}, h^*_{ws}, \theta^*_{ws})$ and $rbox^{ss}(x_{ss}, y_{ss}, w_{ss}, h_{ss}, \theta_{ss})$ predicted by the SS branch is:

$$L_{ss} = \frac{1}{\sum cn^*_{pos}} \sum_{(x^*,y^*)} \mathbb{1}_{\{c^*_{(x^*,y^*)}>0\}} cn^*_{(x^*,y^*)} L_{reg}(rbox^{ws*}_{(x^*,y^*)}, rbox^{ss}_{(x^*,y^*)}) \tag{6}$$

$$L_{reg}(rbox^{ws*}, rbox^{ss}) = \gamma_1 L_{xy} + \gamma_2 L_{wh\theta}, \quad L_{xy} = \sum_{t \in (x,y)} l_1(t^*_{ws}, t_{ss}) \tag{7}$$

$$L_{wh\theta} = \min\{L_{iou}(B_{ws}, B^1_{ss}) + |\sin(\theta^*_{ws} - \theta_{ss})|, L_{iou}(B_{ws}, B^2_{ss}) + |\cos(\theta^*_{ws} - \theta_{ss})|\}$$

where $B_{ws}(-w^*_{ws}, -h^*_{ws}, w^*_{ws}, h^*_{ws})$, $B^1_{ss}(-w_{ss}, -h_{ss}, w_{ss}, h_{ss})$ and $B^2_{ss}(-h_{ss}, -w_{ss}, h_{ss}, w_{ss})$. We set $\gamma_1 = 0.15$ and $\gamma_2 = 1$ by default. $L_{wh\theta}$ takes into account the loss discontinuity caused by the boundary issues (Yang et al., 2021c), such as periodicity of angle and exchangeability of edges.

The overall loss is a weighted sum of the WS loss and the SS loss where we set $\lambda = 0.4$ by default.

$$L_{total} = L_{ws} + \lambda L_{ss} \tag{8}$$

## 4 EXPERIMENTS

### 4.1 DATASETS AND IMPLEMENTATION DETAILS

**DOTA-v1.0** (Xia et al., 2018) is one of the largest datasets for oriented object detection in aerial images, which contains challenging cases, e.g. large-scale dense scenes and complex background. It contains 15 categories, 2,806 images and 188,282 instances with both RBox and HBox annotations, and the latter are directly derived from the former one. The proportion of the training set, validation set, and testing set is 1/2, 1/6, and 1/3, respectively. For training and testing, we follow a standard protocol by cropping images into 1,024×1,024 patches with a stride of 824. **DIOR-R** (Cheng et al., 2022) is an aerial image dataset annotated by RBoxes based on its horizontal annotation version DIOR (Li et al., 2020). There are 23,463 images and 190,288 instances with 20 classes.

Methods are implemented both by PyTorch (Paszke et al., 2019)-based framework MMRotate (Zhou et al., 2022) and Jittor (Hu et al., 2020)-based framework JDet. We adopt the FCOS (Tian et al., 2019) with ResNet50 (He et al., 2016) backbone and FPN neck (Lin et al., 2017a) as the baseline method and building block based on which we develop our approach (see Fig. 1). To implement the weakly-supervised HBox-Mask-RBox alternatives for comparison, we use two strong HBox annotation-based instance segmentation methods: BoxInst and BoxLevelSet, followed by finding its minimum compact surrounding rectangle as the detected RBox and we dub them BoxInst-RBox and BoxLevelSet-RBox respectively. All models are trained with AdamW (Loshchilov & Hutter, 2018) on GeForce RTX 3090 GPU, except BoxLevelSet (Li et al., 2022b) which requires NVIDIA V100 with larger memory. The initial learning rate is $10^{-4}$ with 2 images per mini-batch. The weight decay is 0.05. Besides, we adopt learning rate warm-up for 500 iterations, and the learning rate is divided by 10 at each decay step. Random flipping is adopted without any additional tricks.

Table 1: Results of box the default $AP_{50}$ (%) on the DOTA-v1.0. All models are trained with ResNet50. '1x' and '3x' schedules indicate 12 epochs and 36 epochs for training. $*$ indicates using NV V100 GPU with more memory. MS denotes multi-scale (Zhou et al., 2022) training and testing. See the appendix for performance of specific categories.

| Method | Sched. | MS | Size | Mem. (GB) | FPS | $AP_{50}$ |
|---|---|---|---|---|---|---|
| *RBox-supervised:* | | | | | | |
| RepPoints (Yang et al., 2019b) | 1x | | 1,024 | 3.44 | 24.5 | 64.18 |
| RetinaNet (Lin et al., 2017b) | 1x | | 1,024 | 3.61 | 25.4 | 67.83 |
| RetinaNet (Lin et al., 2017b) | 1x | ✓ | 1,024 | 4.17 | – | 73.30 |
| CSL (Yang & Yan, 2020) | 1x | | 1,024 | 3.93 | 24.6 | 68.26 |
| GWD (Yang et al., 2021c) | 1x | | 1,024 | 3.61 | 25.4 | 69.25 |
| KLD (Yang et al., 2021d) | 1x | | 1,024 | 3.61 | 25.4 | 69.64 |
| KFIoU (Yang et al., 2023b) | 1x | | 1,024 | 3.61 | 25.4 | 70.05 |
| SASM (Hou et al., 2022) | 1x | | 1,024 | 3.69 | 24.4 | 70.35 |
| $R^3$Det (Yang et al., 2021b) | 1x | | 1,024 | 3.78 | 20.0 | 71.17 |
| $S^2$A-Net (Han et al., 2021a) | 1x | | 1,024 | 3.37 | 23.3 | 74.13 |
| FCOS (Tian et al., 2019) | 1x | | 1,024 | 4.66 | 29.5 | 70.78 |
| FCOS (Tian et al., 2019) | 3x | | 1,024 | 4.66 | 29.5 | 72.22 |
| FCOS (Tian et al., 2019) | 1x | ✓ | 1,024 | 6.23 | – | 75.31 |
| *HBox-supervised:* | | | | | | |
| BoxInst-RBox (Tian et al., 2021) | 1x | | 960 | 19.93 | 2.7 | 53.59 |
| BoxLevelSet-RBox$^*$ (Li et al., 2022b) | 1x | | 960 | 26.81 | 4.7 | 56.44 |
| H2RBox (FCOS-based) | 1x | | 960 | 6.25 | 31.6 | 67.90 |
| H2RBox (FCOS-based) | 1x | | 1,024 | 7.02 | 29.1 | 67.82 |
| H2RBox (FCOS-based) | 3x | | 960 | **6.25** | **31.6** | 70.73 |
| H2RBox (FCOS-based) | 3x | | 1,024 | 7.02 | 29.1 | 70.41 |
| H2RBox (FCOS-based) | 1x | ✓ | 1,024 | 8.58 | – | **74.40** |

Table 2: Results of box AP (%) on the DIOR-R `test`. All models are trained with ResNet50. The input image size is 800×800. '1x' and '3x' schedules indicate 12 epochs and 36 epochs. $*$ indicates using NV V100 GPU with more memory.

| Method | Sched. | Mem. (GB) | FPS | AP | $AP_{50}$ | $AP_{75}$ |
|---|---|---|---|---|---|---|
| *RBox-supervised:* | | | | | | |
| RetinaNet (Lin et al., 2017b) | 1x | 2.48 | 33.3 | 33.47 | 54.60 | 33.80 |
| KLD (Yang et al., 2021d) | 1x | 2.48 | 33.3 | 35.77 | 58.00 | 37.00 |
| GWD (Yang et al., 2021c) | 1x | 2.48 | 33.3 | 37.01 | 57.80 | 38.20 |
| FCOS (Tian et al., 2019) | 1x | 3.06 | 40.8 | 34.16 | 58.60 | 31.90 |
| *HBox-supervised:* | | | | | | |
| BoxLevelSet-RBox$^*$ (Li et al., 2022b) | 1x | 11.44 | 4.7 | 29.96 | 56.56 | 24.36 |
| BoxInst-RBox (Tian et al., 2021) | 1x | 9.23 | 3.1 | 31.73 | 57.40 | 28.10 |
| H2RBox (FCOS-based) | 1x | **4.52** | **34.9** | **33.15** | **57.00** | **32.60** |

## 4.2 MAIN RESULTS

**Results on DOTA-v1.0.** As shown in Tab. 1, our method significantly outperforms BoxInst-RBox and BoxLevelSet-RBox by 14.31% and 11.46% in terms of $AP_{50}$, respectively. Moreover, our methods are also more memory and inference efficient. Specifically, compared to BoxInst, we only need less than one-third of its memory (6.25 GB vs. 19.93 GB) and have a about $12\times$ speed advantage (31.6 fps vs. 2.7 fps). In contrast to BoxLevelSet, our memory costs only a quarter of its memory (6.25 GB vs. 26.81 GB), and inference is about 7 times faster (31.6 fps vs. 4.7 fps). In fact, the main cost of the -RBox methods come from the costive post-processing step for find the compact surrounding box as RBox which is fulfilled by calling an OpenCV function in our implementation. Even compared with RBox-supervised methods, our method has outperformed several methods, such as RepPoints and RetinaNet. Under the '1x' and '3x' training schedules, our method slightly lags behind the baseline method, i.e. FCOS (recall it is RBox-supervised), by 2.96% and 1.81%. After using multi-scale training and testing, the gap is reduced to only 0.91% (75.31% vs. 74.40%).

**Results on DIOR-R.** Note that some categories in this dataset including Chimney, Wind mill, Airport, Golf field, are all forcefully annotated by horizontal boxes though the objects are not exactly

Table 3: Ablation for H2RBox with different border effect dismissing strategies for view generation by padding/cropping on DOTA-v1.0.

| Padding | Cropping | AP | $AP_{50}$ | $AP_{75}$ |
|---|---|---|---|---|
| Zeros | | 20.17 | 51.76 | 12.91 |
| Zeros | ✓ | 33.72 | 63.95 | 30.00 |
| Reflection | | **35.92** | **67.31** | **32.78** |
| Reflection | ✓ | 33.60 | 64.09 | 30.02 |

Table 4: Ablation with different label re-assignment strategies. O2M and O2O represent one-to-many and one-to-one.

| Dataset | Assigner | AP | $AP_{50}$ | $AP_{75}$ |
|---|---|---|---|---|
| DOTA | O2M | 21.60 | 53.96 | 14.14 |
| | O2O | **35.92** | **67.31** | **32.78** |
| DIOR-R | O2M | 31.10 | 56.00 | 29.80 |
| | O2O | **33.15** | **57.00** | **32.60** |

Table 5: Ablation with two strategies S1, S2 dealing with circular category: ST & RA on DOTA-v1.0.

| S1 | S2 | ST | RA | AP | $AP_{50}$ | $AP_{75}$ |
|---|---|---|---|---|---|---|
| | | 69.82 | 38.87 | 31.90 | 64.52 | 27.11 |
| ✓ | | 85.29 | 64.04 | 36.36 | 67.25 | 33.26 |
| | ✓ | 84.58 | **65.98** | 35.92 | **67.31** | 32.78 |
| ✓ | ✓ | **85.41** | 63.38 | **36.41** | 67.22 | **33.40** |

Table 6: Ablation with using SS loss ($L_{ss}$) or not on DOTA-v1.0 and DIOR-R.

| Dataset | $L_{ss}$ | AP | $AP_{50}$ | $AP_{75}$ |
|---|---|---|---|---|
| DOTA | | 12.63 | 37.13 | 7.54 |
| | ✓ | **35.92** | **67.31** | **32.78** |
| DIOR-R | | 15.27 | 29.60 | 13.60 |
| | ✓ | **33.15** | **57.00** | **32.60** |

horizontal, which may affect the learning and the final results. As shown in Tab. 2, compared with DOTA-v1.0, DIOR-R is less challenging for the instance segmentation methods. This may explain the observation that the performance of H2RBox and BoxInst-RBox on $AP_{50}$ is close. Yet for high-precision detection i.e. with high $AP_{75}$ that requires more accurate segmentation, H2RBox outperforms BoxLevelSet-RBox and BoxInst-RBox on $AP_{75}$ by 8.24% (32.6% vs. 24.36%) and 4.50% (32.6% vs. 28.10%), and with lower memory and high inference speed. Similarly, H2RBox performs slightly inferior than the RBox-supervised FCOS: 33.15% vs. 34.16%.

## 4.3 ABLATION STUDIES

The ablation study is performed on the proposed H2RBox with 12 training epochs.

**Border effect elimination for view generation.** Tab. 3 studies the impact of different border effect elimination strategies for view generation, in terms of padding and/or cropping (see Sec. 3.1). Such techniques are essential to avoid ground truth angle information leakage, otherwise the model will suffer overfitting and leads to significant performance drop as verified in the first row of the table. Note that when both reflection padding and cropping are applied the AP slightly drops from 35.92% to 33.60% compared with only using reflection padding. The reason may be due to that reduced size of input image by cropping. Hence in all other experiments we always use reflection padding alone.

**Label re-assignment.** Tab. 4 shows the one-to-one strategy outperforms one-to-many strategy.

**Strategies for dealing with sotropic circular object classes.** For circular objects like Storage Tank (ST) and Roundabout (RA), the self-supervised loss takes no effect as it is insensitive to isotropic information. We take two treatments to handle such circular objects. **S1:** for training, we mask the SS loss for circular classes. **S2:** for testing, the horizontal circumscribed rectangle of the circular category is taken as the final output. Tab. 5 shows that, when either or both strategies is used, the performance can be greatly improved, about 15% on ST and about 25% on RA.

**Self-supervised loss.** Without using SS loss, Tab. 6 shows that our method only achieves 12.63% and 15.27% on DOTA-v1.0 and DIOR-R, respectively. In contrast, the use of SS loss leads to a substantial increase in overall performance, reaching 35.92% and 33.15%. Figure 4(b) also shows that the SS loss can effectively help the model learn the correct object angle information.

## 5 CONCLUSION

This paper presents H2RBox, the first (to the best of our knowledge) HBox-supervised oriented object detector. H2RBox learns the rotation via self-supervised learning, whose loss measures the consistency of the predicted angles in two different views. Compared to the alternative HBox-supervised instance segmentation methods, H2RBox achieves much higher detection accuracy especially for complex scenes, yet with lower memory and higher speed. Compared with fully RBox-supervised algorithms, our method still shows competitive.

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

## A  FEASIBLE SOLUTIONS UNDER DIFFERENT CONSTRAINTS

Three different constraints, including horizontal circumscribed rectangle constraint (HCRC), scale constraint (SC) and angle constraint (AC), are introduced in this paper to guide the model to learn the correct result. Fig. 6(a) shows when there are only horizontal circumscribed rectangle constraint, the feasible solutions are still infinite. After adding scale constraint, only the symmetric case and the correct case are left, as shown in Fig. 6(b). The final angle constraint allows the correct solution to be preserved, refer to Fig. 6(c).

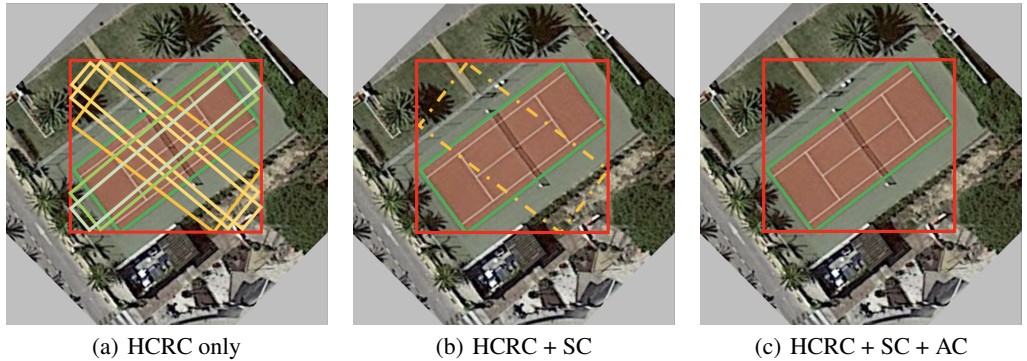

(a) HCRC only                (b) HCRC + SC                (c) HCRC + SC + AC

Figure 6: Visualization of feasible solutions under different constraints.

Table 7: Results of box the default $AP_{50}$ (%) on the DOTA-v1.0. All models are trained with ResNet50. '1x' and '3x' schedules indicate 12 epochs and 36 epochs for training. $^*$ indicates using NV V100 GPU with more memory. MS denotes multi-scale (Zhou et al., 2022) training and testing.

| Method | Sched. | MS | Size | Mem. (GB) | FPS | $AP_{50}$ |
|---|---|---|---|---|---|---|
| *RBox-supervised:* | | | | | | |
| RepPoints (Yang et al., 2019b) | 1x | | 1,024 | 3.44 | 24.5 | 64.18 |
| RetinaNet (Lin et al., 2017b) | 1x | | 1,024 | 3.61 | 25.4 | 67.83 |
| RetinaNet (Lin et al., 2017b) | 1x | ✓ | 1,024 | 4.17 | – | 73.30 |
| CSL (Yang & Yan, 2020) | 1x | | 1,024 | 3.93 | 24.6 | 68.26 |
| GWD (Yang et al., 2021c) | 1x | | 1,024 | 3.61 | 25.4 | 69.25 |
| KLD (Yang et al., 2021d) | 1x | | 1,024 | 3.61 | 25.4 | 69.64 |
| KFIoU (Yang et al., 2023b) | 1x | | 1,024 | 3.61 | 25.4 | 70.05 |
| SASM (Hou et al., 2022) | 1x | | 1,024 | 3.69 | 24.4 | 70.35 |
| $R^3$Det (Yang et al., 2021b) | 1x | | 1,024 | 3.78 | 20.0 | 71.17 |
| ATSS (Zhang et al., 2020) | 1x | | 1,024 | 3.32 | 26.5 | 71.98 |
| $S^2$A-Net (Han et al., 2021a) | 1x | | 1,024 | 3.37 | 23.3 | 74.13 |
| FCOS (Tian et al., 2019) | 1x | | 1,024 | 4.66 | 29.5 | 70.78 |
| FCOS (Tian et al., 2019) | 3x | | 1,024 | 4.66 | 29.5 | 72.22 |
| FCOS (Tian et al., 2019) | 1x | ✓ | 1,024 | 6.23 | – | 75.31 |
| *HBox-supervised:* | | | | | | |
| BoxInst-RBox (Tian et al., 2021) | 1x | | 960 | 19.93 | 2.7 | 53.59 |
| BoxLevelSet-RBox$^*$ (Li et al., 2022b) | 1x | | 960 | 26.81 | 4.7 | 56.44 |
| H2RBox (ATSS-based) | 1x | | 1,024 | 5.50 | 25.7 | 67.24 |
| H2RBox (FCOS-based) | 1x | | 960 | 6.25 | 31.6 | 67.90 |
| H2RBox (FCOS-based) | 1x | | 1,024 | 7.02 | 29.1 | 67.82 |
| H2RBox (FCOS-based) | 3x | | 960 | **6.25** | **31.6** | 70.73 |
| H2RBox (FCOS-based) | 3x | | 1,024 | 7.02 | 29.1 | 70.41 |
| H2RBox (FCOS-based) | 1x | ✓ | 1,024 | 8.58 | – | **74.40** |

Table 8: Results of each category on the DOTA-v1.0 test set.

| Method | PL | BD | BR | GTF | SV | LV | SH | TC | BC | ST | SBF | RA | HA | SP | HC | mAP$_{50}$ |
|---|---|---|---|---|---|---|---|---|---|---|---|---|---|---|---|---|
| *RBox-supervised:* | | | | | | | | | | | | | | | | |
| RepPoints-1x(2019b) | 84.79 | 73.35 | 40.68 | 56.51 | 71.56 | 52.21 | 73.40 | 90.64 | 76.25 | 85.15 | 58.77 | 61.43 | 54.91 | 64.43 | 18.57 | 64.18 |
| RetinaNet-1x (2017b) | 89.11 | 74.52 | 44.69 | 72.18 | 71.80 | 63.59 | 74.94 | 90.78 | 78.71 | 80.56 | 50.48 | 59.17 | 62.86 | 64.35 | 39.69 | 67.83 |
| RetinaNet-1x-ms (2017b) | 88.10 | 84.43 | 50.54 | 79.12 | 73.65 | 59.80 | 72.94 | 90.39 | 86.45 | 87.24 | 65.02 | 65.55 | 67.09 | 70.72 | 58.44 | 73.30 |
| CSL-1x (2020) | 89.03 | 78.25 | 40.04 | 68.52 | 77.20 | 67.14 | 78.25 | 90.87 | 82.77 | 81.30 | 52.17 | 60.33 | 56.14 | 65.71 | 36.10 | 68.26 |
| GWD-1x (2021c) | 88.68 | 78.59 | 45.41 | 71.46 | 72.27 | 68.26 | 77.05 | 90.80 | 80.56 | 81.93 | 46.48 | 60.14 | 63.87 | 67.39 | 46.06 | 69.25 |
| KLD-1x (2021d) | 88.27 | 76.22 | 46.22 | 72.73 | 72.11 | 67.84 | 77.63 | 90.77 | 80.67 | 83.03 | 52.74 | 62.23 | 64.91 | 65.95 | 43.22 | 69.64 |
| KFIoU-1x (2023b) | 88.83 | 77.51 | 47.79 | 74.28 | 71.27 | 62.72 | 74.75 | 90.72 | 82.34 | 81.61 | 58.44 | 64.23 | 64.39 | 67.87 | 44.07 | 70.05 |
| SASM-1x (2022) | 87.44 | 71.31 | 48.46 | 68.07 | 73.93 | 74.24 | 83.55 | 90.91 | 80.36 | 84.59 | 57.98 | 62.84 | 66.51 | 63.82 | 41.17 | 70.35 |
| R$^3$Det-1x (2021b) | 88.96 | 76.99 | 47.09 | 70.89 | 77.54 | 76.19 | 86.24 | 90.91 | 79.45 | 83.60 | 52.98 | 62.50 | 64.65 | 67.32 | 42.29 | 71.17 |
| ATSS-1x (2020) | 88.34 | 76.87 | 50.92 | 71.29 | 76.39 | 76.21 | 83.47 | 90.64 | 81.38 | 83.59 | 58.86 | 60.38 | 65.23 | 67.96 | 48.19 | 71.98 |
| S$^2$A-Net (2021a) | 89.07 | 82.76 | 51.94 | 72.17 | 78.85 | 79.56 | 87.37 | 90.90 | 85.97 | 84.92 | 59.67 | 63.37 | 67.24 | 68.59 | 49.57 | 74.13 |
| FCOS-1x (2019) | 88.41 | 75.61 | 47.98 | 60.10 | 79.78 | 77.81 | 86.64 | 90.08 | 78.23 | 84.95 | 52.80 | 66.25 | 64.45 | 68.28 | 40.31 | 70.78 |
| FCOS-3x (2019) | 88.41 | 76.77 | 49.00 | 59.16 | 79.23 | 79.04 | 86.86 | 90.06 | 75.83 | 83.75 | 58.59 | 59.54 | 69.25 | 72.44 | 53.54 | 72.22 |
| FCOS-1x-ms (2019) | 88.72 | 78.77 | 51.73 | 71.27 | 81.03 | 83.70 | 87.99 | 90.28 | 83.70 | 86.75 | 65.18 | 65.77 | 74.90 | 78.18 | 41.68 | 75.31 |
| *HBox-supervised:* | | | | | | | | | | | | | | | | |
| BoxInst-RBox-1x (2021) | 68.43 | 40.75 | 33.07 | 32.29 | 46.91 | 55.43 | 56.55 | 79.49 | 66.81 | 82.14 | 41.24 | 52.83 | 52.80 | 65.04 | 29.99 | 53.59 |
| BoxLevelSet-RBox*-1x (2022b) | 63.48 | 71.27 | 39.34 | 61.06 | 41.89 | 41.03 | 45.83 | 90.87 | 74.12 | 72.13 | 47.59 | 62.99 | 50.00 | 56.42 | 28.63 | 56.44 |
| H2RBox-ATSS-1x | 87.82 | 74.73 | 43.22 | 69.57 | 72.67 | 53.95 | 70.91 | 90.39 | 85.58 | 83.44 | 54.77 | 63.77 | 47.15 | 66.28 | 44.29 | 67.24 |
| H2RBox-FCOS-1x | 88.47 | 73.51 | 40.81 | 56.89 | 77.48 | 65.42 | 77.87 | 90.88 | 83.19 | 85.27 | 55.27 | 62.90 | 52.41 | 63.63 | 43.26 | 67.82 |
| H2RBox-FCOS-3x | 88.24 | 79.30 | 42.76 | 55.79 | 78.90 | 72.70 | 77.54 | 90.85 | 81.96 | 84.38 | 55.28 | 64.49 | 61.91 | 70.63 | 51.51 | 70.41 |
| H2RBox-FCOS-1x-ms | 88.93 | 78.89 | 46.27 | 68.79 | 81.12 | 75.45 | 86.68 | 90.89 | 86.71 | 87.33 | 64.15 | 68.83 | 62.81 | 69.39 | 59.79 | 74.40 |

# B  VERIFICATION EXPERIMENTS ON DIFFERENT DETECTORS

As shown in Tab. 7, we have conducted experiments on different basic detectors, including anchor based method (ATSS (Zhang et al., 2020)) and anchor free method (FCOS (Tian et al., 2019)). It can be seen that the method proposed in this paper has excellent portability. Tab. 8 lists the performance of each class of each method in Tab. 7.

