# OpenReview forum: "H2RBox: Horizontal Box Annotation is All You Need for Oriented Object Detection"
_ICLR.cc/2023/Conference — ICLR 2023 poster_

### Official Review · Reviewer_jpAe · 2022-10-19

**Confidence:** 4
**Correctness:** 4
**Technical Novelty And Significance:** 3
**Empirical Novelty And Significance:** 3
**Recommendation:** 8

**Clarity, Quality, Novelty And Reproducibility:**

The writting of this manuscript is okay. The proposed method is novel. The reproducibility is okay if the code is released.

**Strength And Weaknesses:**

## Strength:
### (1) The proposed method is interesting since it can predict the angle of the object without the supervision of the angle.
### (2) The experimental results show that it can achieve competing results compared to the fully-supervised methods.

## Weaknesses:
### (1) The proposed method need more GPU memory than previous methods, as listed in Table 1.
### (2) The definition of the angles of the objects are quite fusing for me. For example, a horizontal box that exactly match the object can be either 0 degree or 90 degree. How to distinguish them?
### (3) The contributions listed in the Introduction Section are weak since two items of them are the experimental results. It would be better to specify the technical contributions instead of the experimental results.

**Summary Of The Paper:**

This paper proposed an oriented object detector that used horizontal box annotations. The proposed method applied weakly- and self-supervised learning for predicting the angle of the object. The experimental results on standard benchmarks show the effectiveness of the proposed method.

**Summary Of The Review:**

Considering the strength and weaknesses of this paper, I tend to give a positve recommendation.

---

> ### Author Response · Authors · 2022-11-10
> **Response to Reviewer jpAe (Round 1)**
>
> > ***Q1: The proposed method need more GPU memory than previous methods, as listed in Table 1.***
>
> **A1:** Thanks for your comment. Table 1 lists the GPU memory required only for training. Since we use the Siamese network, the memory required is slightly more than RBox-supervised method, but already much less than HBox supervised instance segmentation method, e.g. BoxInst and BoxLevelSet. Furthermore, in the inference phase, we only use weakly supervised branch (WS), so both memory and speed are similar to RBox-supervised method.
>
> In summary, HBox labeling cost much less manual efforts while leads to a bit higher memory cost in training which we believe is not an issue in practice.
>
>
> > ***Q2: The definition of the angles of the objects are quite fusing for me. For example, a horizontal box that exactly match the object can be either 0 degree or 90 degree. How to distinguish them?.***
>
> **A2:** The open source benchmark, i.e. MMRotate, gives angle representations under different definitions, please refer to: https://mmrotate.readthedocs.io/en/latest/intro.html
>
> Specifically:
> - **For OpenCV definition, $\theta \in (0^\circ,90^\circ]$**:
>
> $(x,y,w_{oc},h_{oc}, 90^\circ)$. Example: AB=10, AD=4, Box: $(5,2,4,10,90^\circ)$
> ```
> 0-------------------> x
> |  A-------------B
> |  |             |
> |  |     box    woc
> |  |             |
> |  D-----hoc-----C
> v
>
> ```
> $(x,y,w_{oc},h_{oc}, 90^\circ)$. Example: AB=10, AD=4, Box: $(2,5,10,4,90^\circ)$
> ```
> 0-------------------> x
> |  D----------A
> |  |          |
> |  |          |
> |  |          |
> |  |   box   woc
> |  |          |
> |  |          |
> |  |          |
> |  C---hoc----B
> v
> ```
>
> - **For Long Edge definition, $\theta \in [-90^\circ,90^\circ)$：**
>
> $(x,y,w_{le},h_{le}, 0^\circ)$. Example: AB=10, AD=4, Box: $(5,2,10,4,0^\circ)$
> ```
> ^
> |  A-------------B
> |  |             |
> |  |     box    hle
> |  |             |
> |  D-----wle-----C
> 0-------------------> x
> ```
>
> $(x,y,w_{le},h_{le}, -90^\circ)$. Example: AB=10, AD=4, Box: $(2,5,10,4,-90^\circ)$
> ```
> ^
> |  D----------A
> |  |          |
> |  |          |
> |  |          |
> |  |   box   wle
> |  |          |
> |  |          |
> |  |          |
> |  C---hle----B
> 0-------------------> x
> ```
>
>
> > ***Q3: The contributions listed in the Introduction Section are weak since two items of them are the experimental results. It would be better to specify the technical contributions instead of the experimental results.***
>
> **A3:** Thank you for your suggestion, we have revised the corresponding part (marked in blue). Specifically:
>
> **1)** To our best knowledge,  we propose the first HBox annotation-based oriented object detector. Specifically, a weakly- and self-supervised angle learning paradigm is devised which closes the gap between HBox training and RBox testing, and it can serve as a plugin for existing detectors.
>
> **2)** By geometric derivations, we show that the predicted RBox is the correct GT RBox under our designed pipeline and consistency loss, and does not rely on some not-fully-verified/ad-hoc assumptions, e.g. color-pairwise affinity in BoxInst or additional intermediate results whose quality itself is hard to ensure, e.g. feature map used by many weakly supervised methods.
>
> **3)** Compared with the potential alternatives e.g. HBox-Mask-RBox whose instance segmentation part is fulfilled by the state-of-the-art BoxInst, our H2RBox outperforms by about 14\% mAP (67.90\% vs. 53.59\%) on DOTA-v1.0 dataset, requiring only one third of its computational resources (6.25 GB vs. 19.93 GB), and being around 12$\times$ faster in inference (31.6 fps vs. 2.7 fps).
>
> **4)** Compared with the fully RBox annotation-supervised rotation detector FCOS, H2RBox is only 0.91\% (74.40\% vs. 75.31\%) behind on DOTA-v1.0, and even surpasses it by 1.7\% (34.90\% vs. 33.20\%) on DIOR-R. Furthermore, we do not add extra computation in the inference stage, thus maintaining a comparable detection speed, about 29.1 FPS vs. 29.5 FPS on DOTA-v1.0.

---

> > ### Author Response · Authors · 2022-11-22
> > **Update source code.**
> >
> > > ***Q4: The writting of this manuscript is okay. The proposed method is novel. The reproducibility is okay if the code is released.***
> >
> > **A4:** Thanks for your kind review and we feel encouraged. The clean source code is uploaded in the Supplementary Material, you can check it.

---

### Official Review · Reviewer_uvAp · 2022-10-21

**Confidence:** 5
**Clarity, Quality, Novelty And Reproducibility:** This paper is well originated with in…
**Correctness:** 4
**Technical Novelty And Significance:** 3
**Empirical Novelty And Significance:** 3
**Recommendation:** 6

**Strength And Weaknesses:**

Strength

-The main contribution that building the self-supervised learning strategy on a rotated image is interesting.

-The experiments are sufficient to demonstrate the effectiveness of each component and the overall performance.

Weaknesses

-The authors mainly evaluate the proposed framework on FCOS, it's suggested to verify the model's generalization on other baseline methods for general object detection or oriented object detection.

-The authors used several loss items with empirical trade-offs. It's necessary to explain the reasons and show their effects on final performance.

**Summary Of The Paper:**

This paper proposed a weakly supervised method for oriented object detection by leveraging only horizontal bounding boxes. The network is able to learn to predict the rotated boxes by building the transition functions between a WS branch on original images and a SS branch  on rotated images. By building upon the FCOS detector, the proposed method performs favorably against state-of-the-art weakly supervised approaches.

**Summary Of The Review:**

This paper brings some new insights for weakly supervised oriented object detection. Currently, I'm glad to accept it.

---

> ### Author Response · Authors · 2022-11-10
> **Response to Reviewer uvAp (Round 1)**
>
> > ***Q1: The authors mainly evaluate the proposed framework on FCOS, it's suggested to verify the model's generalization on other baseline methods for general object detection or oriented object detection.***
>
> **A1:** Thank you for your insightful advice. We have conducted experiments with anchor-based method, i.e. ATSS (compared with the anchor free method FCOS in this paper), and the experimental results are shown in the following Table. Due to the limited time, we do not adjust the parameters carefully, but it is sufficient to verify the portability of the proposed method. We believe that the performance will be further improved. Therefore, we will temporarily add these experiments to the appendix of the paper until it is adjusted to a satisfactory result. At the same time, we will also open source the code and have uploaded our clean source code in the Supplementary Material (you can check it).
>
> | Dataset        | Method              | Annotation | mAP   |
> | -------------- | ------------------- | ---------- | ----- |
> | DOTA-v1.0 test | ATSS (anchor-based) | RBox       | 71.98 |
> | DOTA-v1.0 test | H2RBox+ATSS         | HBox       | 67.24 |
> | DOTA-v1.0 test | FCOS (anchor-free)  | RBox       | 70.78 |
> | DOTA-v1.0 test | H2RBox+FCOS         | HBox       | 67.82 |
>
> > ***Q2: The authors used several loss items with empirical trade-offs. It's necessary to explain the reasons and show their effects on final performance.***
>
> **A2:** We empirically found that our loss is not sensitive to the weights when we put them in a blacnced way which is what we set in our experiments. We think the reason is that the parts are basically equally important and all are necessary.
>
> Specifically, the weight of most loss items is set to 1 by default, except the weight of center point loss $L_{xy}$ in Eq. 7. This part has been implicitly included in the weak supervised branch ($L_{ws}$), so we set a smaller weight ($\gamma_{1}=0.15$) in the self supervised branch. Through the experiment, we find that even if it is set to 0, we can still achieve similar performance, but if its value is set large, the training will be unstable. This instability may be caused by overemphasizing this part, which affects the learning of other parts, e.g. scale consistency.

---

> > ### Author Response · Authors · 2022-11-29
> > **Q1/A1 update**
> >
> > More results on more challenging DOTA-v1.5 and DOTA-v2.0, which contain more tiny objects (less than 10 pixels).
> >
> > | Dataset        | Method              | Annotation | mAP   |
> > | -------------- | ------------------- | ---------- | ----- |
> > | DOTA-v1.5 test | ATSS (anchor-based) | RBox       | 63.50 |
> > | DOTA-v1.5 test | H2RBox+ATSS         | HBox       | 59.02 |
> > | DOTA-v1.5 test | FCOS (anchor-free)  | RBox       | 62.24 |
> > | DOTA-v1.5 test | H2RBox+FCOS         | HBox       | 60.19 |
> >
> > | Dataset        | Method              | Annotation | mAP   |
> > | -------------- | ------------------- | ---------- | ----- |
> > | DOTA-v2.0 test | ATSS (anchor-based) | RBox       | 48.57 |
> > | DOTA-v2.0 test | H2RBox+ATSS         | HBox       | 45.35 |
> > | DOTA-v2.0 test | FCOS (anchor-free)  | RBox       | 49.24 |
> > | DOTA-v2.0 test | H2RBox+FCOS         | HBox       | 47.08 |

---

### Official Review · Reviewer_o5T8 · 2022-10-23

**Confidence:** 3
**Clarity, Quality, Novelty And Reproducibility:** In general the quality of this paper …
**Correctness:** 3
**Technical Novelty And Significance:** 2
**Empirical Novelty And Significance:** 3
**Recommendation:** 6

**Strength And Weaknesses:**

Strength
1. The direction is interesting and new.
2. The proposed method achieved very good performance (comapred to the previous methods and oriented box supervised models).

Weakness
1. The method is not very novel. For example using the horizontal circumscribed rectangles as supervision is similar to the techinical used in BoxInst[1]. The scale consistency loss is also widely used in previous methods, such as [2, 3].

[1] https://arxiv.org/abs/2012.02310
[2] https://arxiv.org/abs/1905.02249
[3] https://arxiv.org/pdf/2004.04581.pdf

**Summary Of The Paper:**

This paper proposed a method to detect oriented object with only horizontal bounding box annotations. In detail, it designed two losses. The first is the weakly supervised loss and the second is the self-supervised consistency loss. The proposed method outperformed the box supervised instance segmentation method such as BoxInst. It is also close to the oriented box supervised model.

**Summary Of The Review:**

The problem studied in this paper is new. But the method is not very novel. But the performance is good. So I prefer to accept this paper.

---

> ### Author Response · Authors · 2022-11-10
> **Response to Reviewer o5T8 (Round 1)**
>
> > ***Q1: The method is not very novel. For example using the horizontal circumscribed rectangles as supervision is similar to the techinical used in BoxInst [1]. The scale consistency loss is also widely used in previous methods, such as [2, 3].***
>
> **A1:** Thank you for your comment.
>
> We emphasize that the main novelty is not the individual components which can be found in isolation in other works as mentioned by the reviewer. While as a whole, we belive our work contributes the community by defining the new task setting, and a more practical and cost-saving approach for low-cost training of rotation detectors, especially as the first HBox annotation-based oriented object detector.
>
>
> Although some components may appear in similar form in other different tasks, we believe that they play a different role, which does not affect our innovation and our rationale behind the paper.
>
> [1] https://arxiv.org/abs/2012.02310
> [2] https://arxiv.org/abs/1905.02249
> [3] https://arxiv.org/pdf/2004.04581

---

### Official Review · Reviewer_upuU · 2022-10-27

**Confidence:** 5
**Correctness:** 3
**Technical Novelty And Significance:** 4
**Empirical Novelty And Significance:** 4
**Recommendation:** 10

**Clarity, Quality, Novelty And Reproducibility:**

---
In equ. 7, what are t^{*}_{ws} and t_{ss} denoted?

---
 It is still unclear why the combination of wsl and ssl can figure out the oriented boxes.
This is my main concern.
The wsl branch may predict a lot of boxes with the same horizontal circumscribed rectangle, and the ssl branch tries to learn the same boxes as wsl with random rotation transformation.

**Strength And Weaknesses:**

Compared to the alternative HBox-supervised instance segmentation methods, H2RBox achieves much higher detection accuracy especially for complex scenes, yet with lower memory and higher speed.
Compared with fully RBox-supervised algorithms, our method still shows competitive, and sometimes even better performance.

**Summary Of The Paper:**

This paper presents HBox-supervised oriented object detector.
H2RBox learns the rotation via self-supervised learning, whose loss measures the consistency of the predicted angles in two different views.


**Summary Of The Review:**

This paper has significant contribution to the community, and the results are quite good (very close performance and speed compared to rotated box-supervised methods).
 It may reveal that we can use HBox annotations than RBox one to learn oriented object detectors.

---

> ### Author Response · Authors · 2022-11-10
> **Response to Reviewer upuU (Round 1)**
>
> Thanks for your kind review and we feel encouraged.
>
> >***Q1: In equ. 7, what are $t_{ws}^{\*}$ and $t_{ss}$ denoted?***
>
> **A1:** Thank you for your comment. $t_{ws}^{*}$ and $t_{ss}$ in $L_{xy}=\sum_{t \in (x,y)}l_{1}(t_{ws}^{\*}, t_{ss})$ denote the coordinates of the center point, i.e. ($x_{ws}^{\*}, y_{ws}^{\*}$) and $(x_{ss}, y_{ss})$, predicted in the weak supervised branch and the self supervised branch respectively.
>
> > ***Q2: It is still unclear why the combination of wsl and ssl can figure out the oriented boxes. This is my main concern. The wsl branch may predict a lot of boxes with the same horizontal circumscribed rectangle, and the ssl branch tries to learn the same boxes as wsl with random rotation transformation.***
>
> **A2:** The proof has been given in Section 3.3. Specifically, according to Figure 5 in the paper, we can list the following five equations:
>
> $w_{1} \cdot |\cos\theta| +h_{1} \cdot |\sin\theta| = w_{ws}$,
> $w_{1} \cdot |\sin\theta| +h_{1} \cdot |\cos\theta| = h_{ws}$,
> $w_{2} \cdot |\cos\varphi| +h_{2} \cdot |\sin\varphi| = w_{ss}$,
> $w_{2} \cdot |\sin\varphi| +h_{2} \cdot |\cos\varphi| = h_{ss}$,
> $\varphi-\theta=\Delta\theta$
>
> - First, the weak supervised branch (WS) introduces the horizontal circumscribed rectangle constraint, making $(w_{ws}, h_{ws})$ and $(w_{ss}, h_{s})$ known quantities.
> - Second, $\Delta \theta$ can be determined according to the rotation transformation $R$ of View 2.
> - Third, the scale loss $L_{wh}$ of the self supervised branch (SS) makes the two branches predict the same scale, that is, $w_{1}=w_{2}=w$, $h_{1}=h_{2}=h$
>
> Thus, we have:
>
> $w \cdot |\cos\theta| + h \cdot |\sin\theta| = w_{ws}$,
> $w \cdot |\sin\theta| + h \cdot |\cos\theta| = h_{ws}$,
> $w \cdot |\cos\varphi| +h \cdot |\sin\varphi| = w_{ss}$,
> $w \cdot |\sin\varphi| +h \cdot |\cos\varphi| = h_{ss}$,
> $\varphi-\theta=\Delta\theta$
>
> So far, there are only four variables left ($w,h,\varphi,\theta$), and five equations can get the only solution, that is, the unlabeled GT RBox.

---

### Author Response · Authors · 2022-11-18
**Comments to everyone.**

Dear Reviewers,

Approaching the pdf updating ddl, is there anything needing added.

Best,

Paper116 Authors

---

### Decision · Program_Chairs · 2023-01-20

**Decision:**

Accept: poster

**Justification For Why Not Higher Score:**

The reviewers share the comments that this paper has limited novelty. It does not provide new method or model component design. Instead, it integrates existing components/methods into a working-well framework. This framework is valuable for the community. But the scope of this work and the limited technical insight limits its impact for the community.

**Justification For Why Not Lower Score:**

The framework proposed by this paper works well. It is reasonable and valuable for the community. All the reviewers agree with accept. The reviewers raise some questions about this work and the authors address them well in their response.

**Metareview: Summary, Strengths And Weaknesses:**

This paper studies the oriented object detection. Different from existing works, it explores a weakly-supervised setting where only horizontal bounding box annotations are provided. To address this problem, it introduces a new method that leverages self-supervised learning method (enforcing the orientation prediction consistency of the object with different views). Experiments demonstrate the proposed method outperforms existing works significantly.

Strength:

- This paper explores a useful setting for oriented object detection.
- The proposed method is reasonable and easy to follow.
- The proposed method performs quite well in the experiments.

Weakness:
- The novelty is kind of limited in the aspect that this work does not introduce new method/components.

**Note From Pc:**

if the above contains the word "oral" or "spotlight" please see: "oral" presentation means -> notable-top-5% and "spotlight" means -> notable-top-25%. As stated in our emails, we are disassociating presentation type from AC recommendations